# Basic Properties of Adipose-Derived Mesenchymal Stem Cells of Rheumatoid Arthritis and Osteoarthritis Patients

**DOI:** 10.3390/pharmaceutics15031003

**Published:** 2023-03-20

**Authors:** Ewa Kuca-Warnawin, Weronika Kurowska, Magdalena Plebańczyk, Anna Wajda, Anna Kornatka, Tomasz Burakowski, Iwona Janicka, Piotr Syrówka, Urszula Skalska

**Affiliations:** 1Department of Pathophysiology and Immunology, National Institute of Geriatrics, Rheumatology and Rehabilitation, 02-637 Warsaw, Poland; 2Department of Molecular Biology, National Institute of Geriatrics, Rheumatology and Rehabilitation, 02-637 Warsaw, Poland; 3Rheumaorthopedics Clinic and Polyclinic, National Institute of Geriatrics, Rheumatology and Rehabilitation, 02-637 Warsaw, Poland

**Keywords:** adipose-derived stem cells, rheumatoid arthritis, osteoarthritis, tissue regeneration, immunomodulation

## Abstract

Rheumatoid arthritis (RA) and osteoarthritis (OA) are destructive joint diseases, the development of which are associated with the expansion of pathogenic T lymphocytes. Mesenchymal stem cells may be an attractive therapeutic option for patients with RA or OA due to the regenerative and immunomodulatory abilities of these cells. The infrapatellar fat pad (IFP) is a rich and easily available source of mesenchymal stem cells (adipose-derived stem cells, ASCs). However, the phenotypic, potential and immunomodulatory properties of ASCs have not been fully characterised. We aimed to evaluate the phenotype, regenerative potential and effects of IFP-derived ASCs from RA and OA patients on CD4+ T cell proliferation. The MSC phenotype was assessed using flow cytometry. The multipotency of MSCs was evaluated on the basis of their ability to differentiate into adipocytes, chondrocytes and osteoblasts. The immunomodulatory activities of MSCs were examined in co-cultures with sorted CD4+ T cells or peripheral blood mononuclear cells. The concentrations of soluble factors involved in ASC-dependent immunomodulatory activities were assessed in co-culture supernatants using ELISA. We found that ASCs with PPIs from RA and OA patients maintain the ability to differentiate into adipocytes, chondrocytes and osteoblasts. ASCs from RA and OA patients also showed a similar phenotype and comparable abilities to inhibit CD4+ T cell proliferation, which was dependent on the induction of soluble factors The results of our study constitute the basis for further research on the therapeutic potential of ASCs in the treatment of patients with RA and OA.

## 1. Introduction

Rheumatoid arthritis is a chronic, incurable, autoimmune disease affecting about 1% of the world population, mainly women [1]. The disease is caused by loss of immunological self-tolerance and is characterised by chronic inflammation and progressive destruction of joints [2]. The aetiology of RA is unknown, and many factors contribute to its development, i.e., genetic, environmental and immunological. The classic hallmark of RA pathology is synovial membrane inflammation (synovitis) and hyperplasia. Synovial membrane-building cells—fibroblast-like synoviocytes (FLS)—secrete excessive amounts of proinflammatory cytokines, chemokines and enzymes, degrading joint cartilage and bone. Moreover, FLSs proliferate excessively and are resistant to apoptosis [3]. The immunological response in rheumatoid joint is abnormal due to the imbalance between immunoactivation and immunosuppression.

Osteoarthritis is the most common joint disease. Concepts concerning the pathogenesis of OA have evolved from one originally addressing only cartilage, to a more complex version, involving the entire joint structure. Many risk factors have been identified, of which age and overweight appear to be the most important. Increasing importance is being given to the involvement of low-grade systemic inflammation in the pathogenesis of this disease [4,5,6].

Mesenchymal stem cells (MSCs) are multipotent cells of mesodermal origin. MSCs are found in bone marrow, adipose tissue, Wharton’s jelly, periosteum, tendon, cartilage and synovial membrane [7]. MSCs derived from adipose tissue (adipose-derived mesenchymal stem cells—ASCs) are of special interest in terms of therapeutic application because of their easy, non-invasive isolation (liposuction) and abundant number (500× more than in bone marrow) [8]. These cells can differentiate in vivo into cells of the same embryonal origin, e.g., chondrocytes, osteoblasts, adipocytes, myocytes and cardiomyocytes [9,10]. Moreover, in vitro, MSCs can also differentiate into cells of ecto- and endodermal origin (e.g., epithelial cells, or hepatocytes, respectively) [11,12]. Apart from regenerative potential, MSCs have immunosuppressive properties. MSCs act via the paracrine pathway, microvesicles secretion or by direct cell-to-cell contact with responder cells [13,14]. MSCs have reduced immunogenicity due to the low expression of MHC I and the lack of expression of MHC II and co-stimulatory molecules (CD40, CD80, CD86) [15,16], which makes them very valuable as allogeneic cell transplants.

The proliferation and differentiation of MSCs depend on their niche and substrate, which are the factors that influence the behaviour of cells in their surrounding microenvironment. As the articular cartilage is not supplied with blood, the oxygen concentration in the knee joint is low. The inflammatory process can affect cellular niches. It has been shown that hypoxia is increased in the inflamed joint [17]. Both hypoxia and inflammation may influence the differentiation potential of ASCs obtained from the IFP. For example, studies have shown that hypoxia can enhance the differentiation of MSCs into certain cell types. In a low-oxygen environment, MSCs have been shown to differentiate more readily into bone and cartilage cells. On the other hand, hypoxia has been shown to inhibit the differentiation of MSCs into other cell types, such as adipocytes [18,19,20].

Regarding immunosuppressive and regenerative properties of MSCs, the administration of these cells in patients suffering from autoimmune, inflammatory and degenerative diseases has emerged as a promising treatment strategy. So far, encouraging results of autologous or allogeneic MSC transplants have been obtained in various autoimmune diseases and in the reconstruction of bone, cartilage or soft tissues [11,21,22,23,24]. The potential of MSCs’ therapeutic use is also studied in rheumatoid arthritis (RA).

The aim of our study was to investigate the potential therapeutic use of tissue that is waste from knee replacement surgery. We hypothesise that ASCs localised in the site of ongoing inflammatory process (rheumatoid joint) have impaired regenerative and immunomodulatory properties and do not limit rheumatoid T cell proliferation.

## 2. Materials and Methods

### 2.1. Patients

Patients who fulfilled either the ACR/EULAR criteria for RA (*n* = 15) or the criteria for OA (*n* = 15) were included in the study [25,26]. This study meets all criteria contained in the Declaration of Helsinki and was approved by the Ethics Committee of the National Institute of Geriatrics, Rheumatology, and Rehabilitation, Warsaw, Poland (approval number DL/14.01.2016) All patients gave their written informed consent prior to enrolment. Demographic and clinical data of the patients are shown in Table 1.

### 2.2. ASC Isolation and Culture

Specimens of infrapatellar fat pad were taken from the patients undergoing total knee replacement. The infrapatellar fat pad, also known as Hoffa body, is a small, cylinder-like piece of adipose tissue located beneath the patella in front of the knee joint. ASC isolation and culture were performed as described previously [27]. All experiments were performed using ASCs at 3–5 passages. The medium used for ASC culture was purchased from Lonza (Lonza Group, Lonza Walkershille Inc., Walkersville, MD, USA).

### 2.3. Flow Cytometry Analysis of ASCs Phenotype

For ASC phenotype analysis, cells were detached with non-enzymatic cell dissociation solution (ATCC Manassas, VA, USA). In the next step, ASCs were washed with FACS buffer (phosphate-buffered saline, 0.1% NaN_3_, 1% FCS). Then, 5 × 10^4^ cells were suspended in 50 μL of FACS buffer and stained with antibodies against the following surface markers: CD90-FITC, CD105-PE, CD73-APC (eBioscience, San Diego, CA, USA), CD34-PE-Cy7, CD45-PE, CD19-PE, and CD14-APC (BD Pharmingen, San Diego, CA, USA). After the washing step, cells were acquired and analysed using a FACSCanto cell sorter/cytometer and Diva software. The gating strategy is shown in Appendix A in the Appendix A.

### 2.4. Adipogenic Differentiation of ASCs

ASCs were seeded into a 12-well culture plate at 35 × 10^5^ per well in DMEM/F12/10% FCS culture medium. Cells were cultured in order to obtain 100% of confluence. After this time, the previous medium was harvested, the cells were washed with PBS buffer, and complete culture medium was added to induce adipogenesis (Mesenchymal Stem Cell Adipogenic Differentiation Medium, Cat No 7541, ScienCell, Carlsbad, CA, USA). Cultures were carried out for 21 days. The culture medium was replaced every 3–4 days (except for the negative control where the medium was replaced every 2 days). After completion of culture, cells were stained with Oil Red O (Sigma-Aldrich, St. Louis, MO, USA) to identify adipocytes and subjected to RNA isolation to analyse expression of relevant genes involved in adipogenesis.

In order to stain oil droplets with Oil Red O, cells were fixed for 15 min in 4% formaldehyde at room temperature, then washed with PBS buffer. The wells were flooded with a 2.1% solution of Oil Red O. The plates were then incubated with gentle shaking in a humid chamber for 30 min. Finally, the cells were washed with deionised water, and pictures were taken to compare the staining effect. In order to more accurately estimate the differences between the individual variants, after staining, dye was extracted with isopropanol. The amount of dye was determined spectrophotometrically at a wavelength of 510 nm, in the Multiskan GO spectrophotometer (Thermo Fischer Scientific, Waltham, MA, USA) using Skanit Software 3.2 (Thermo Fischer Scientific) for analysis.

### 2.5. Osteogenic Differentiation of ASCs

ASCs were seeded into a 12-well culture plate at 11 × 10^5^ per well in DMEM/F12/10% FCS culture medium. Cells were cultured for 24 h to adhere to the surface of the plate. After this time, the previous medium was harvested, the cells were washed with PBS buffer, and complete culture medium was added to induce osteogenesis (Mesenchymal Stem Cell Osteogenic Differentiation Medium, Cat No 7531, ScienCell, Carlsbad, CA, USA). Cultures were maintained for 21 days. The culture medium was replaced every 3–4 days (except for the negative control where the medium was replaced every 2 days). After completion of culture, cells were stained with Alizarin Red S (Alizarin Red S Staining Kit ScienCell, Carlsbad, CA, USA) and subjected to RNA isolation to analyse expression of relevant genes involved in osteogenesis.

In order to stain calcium deposits with Alizarin Red S, cells were fixed for 10 min in 4% formaldehyde at room temperature, then washed with PBS buffer. The wells were flooded with a 1% aqueous solution of Alizarin Red S, pH 6.4. The plates were then incubated in a humid chamber for 1 h 20 min. Finally, the cells were washed with PBS and pictures were taken to compare the staining effect. In order to more accurately estimate the differences between the individual variants, after staining, dye was extracted. The amount of dye was determined spectrophotometrically at a wavelength of 450 nm, in the Multiskan GO spectrophotometer (Thermo Fischer Scientific) using Skanit Software 3.2 (Thermo Fischer Scientific) for analysis.

### 2.6. Chondrogenic Differentiation of ASCs

Suspension of 0.5–1 × 10^6^ undifferentiated ASCs in DMEM/F12/10% FCS culture medium was placed in a 15 mL tube. The suspension was centrifuged at room temperature for 5 min at 250 g. The supernatant was removed, followed by the addition of 1 mL of complete chondrogenesis induction medium (Mesenchymal Stem Cell Chondrogenic Differentiation Medium, Cat No 7551, ScienCell, Carlsbad, CA, USA), which contained TGF-β1 at a concentration of 10 ng/mL. During this and following steps it was essential that the cell pellet remained intact which ensured interactions between cells crucial for the chondrogenesis process. The chondrogenic culture was maintained for 28 days. The culture medium was replaced every 2–3 days. The caps of the test tubes were loosened to provide the cells with access to oxygen. After completion of the chondrogenesis, the pellets were subjected to RNA isolation to analyse expression of genes involved in chondrogenesis.

### 2.7. Gene Expression Analysis

At the end of the cultures, total RNA was isolated from the cells. TRIzol solution (for pellets after homogenisation) was added to the cells. After 5 min of incubation, the samples were frozen at −70 °C. After thawing, the samples were shaken with chloroform. The resulting RNA-containing aqueous phase was purified using the RNAeasy kit (Qiagen, Hilden, Germany) according to the manufacturer’s instructions. The purity and concentration of the isolated RNA was determined using the Multiskan GO spectrophotometer (Thermo Fischer Scientific) and the Skanit Software 3.2 analysis software (Thermo Fischer Scientific). 

For the reverse-transcription polymerase chain reaction (RT-PCR), the cDNA High-Capacity cDNA Reverse Transcription Kit (Thermo Fischer Scientific) was used. Samples of 10 ng of total RNA were used to perform the reverse-transcription. Samples were incubated for 10 min at 25 °C, then for 120 min at 37 °C, followed by 10 min at 85 °C. Finally, samples were cooled to 4 °C. The reverse-transcription reaction was performed in a Biometra Tgradient thermocycler (Analytik Jena, Jena, Germany). 

The PCR reaction was performed in a volume of 10 µL that included 2 µL of RT product, 5 µL of TaqMan Gene Expression Master Mix, 0.5 µL of probe mix (TaqMan Thermo Fisher Scientific, Waltham, MA, USA), and 2.5 µL of water (Genoplast, Rokocin, Poland). Used probes are shown in Table 2. Reactions were performed at 50 °C for 2 min, 95 °C for 10 min, followed by 50 cycles at 95 °C for 15 s, and 60 °C for 1 min. Samples were analysed in triplicate using the QuantStudio 5 RT-qPCR machines (Thermo Fisher Scientific, Waltham, MA, USA). Data were analysed in Quant Studio Design & Analysis Software 1.3.1 (Thermo Fisher Scientific, Waltham, MA, USA). Gene expression was evaluated using the ΔΔCT-method. Values of CT ≥ 35 were treated as below quantification. The most stable reference genes have been selected for adipogenesis (GUSB and EEF1) for osteogenesis and chondrogenesis (TBP and RPL13a).

### 2.8. Co-Cultures of ASCs with Purified Allogeneic CD4+ T-Cells or PBMCs and Proliferation Assay

A total of 6 × 10^4^ ASCs were seeded on each well of 24-well plates. Both untreated and cytokine-pre-stimulated ASCs were used in the experiments. For pre-stimulation, ASCs were treated for 24 h with recombinant human tumour necrosis factor (TNF) and interferon (IFN-γ) (both from R&D Systems, Minneapolis, MN, USA; each used at 10 ng/mL).

Peripheral blood mononuclear cells (PBMCs) were isolated from buffy coats by density gradient centrifugation with Ficoll-Paque (GE Healthcare, Uppsala, Sweden). PBMCs were stained with cell trace violet (CTV) (Thermo Fisher Scientific) and stimulated with 2.5 μg/mL of PHA (Sigma Aldrich). Activated PBMCs were co-cultured with ASCs (1.2 × 10^6^ PBMCs/well/2 mL of medium).

The CD3+CD4+ cells were obtained from PBMCs using magnetic separation with the EasySep Human CD4+ T-Cell Isolation Kit (Stemcell Technologies, Vancouver, Canada). In the next step, purified CD3+CD4+ T lymphocytes or PBMCs were stained with CTV. Stained CD4+ T-cells were activated with Dynabeads Human T-Activator CD3/CD28 (Thermo Fisher Scientific) and then co-cultured with ASCs (1.2 × 10^6^ CD4+ T cells/well/2 mL of medium). After 5 days of co-culture, CD4+ T-cells or PBMCs were harvested for cytometric proliferation analysis. In order to achieve this, the percentage of proliferating cells and proliferation index (PI—number of divisions per proliferating cell) were calculated as shown before [28]. The gating strategy is shown in Appendix A in the Appendix A.

### 2.9. Determination of the Concentration of Soluble Factors Involved in the Antiproliferative Properties of ASCs

In culture supernatants obtained at the end of ASCs and PBMCs co-culture, the concentrations of substances involved in the antiproliferative effects of ASCs were determined. IL-10 and TGFb concentrations were determined using ELISA assays (Thermo Fischer Scientifici), while PGE_2_ concentrations were determined using a competence ELISA (R&D, Biotechne). Kynurenine production was determined using the method described previously [29].

### 2.10. Blocking Experiments

To further investigate soluble factors in the immunomodulatory capacity of ASCs, experiments were performed to block the action of these factors in ASCs/PBMCS co-culture. Neutralising antibodies were used to neutralise the effect of cytokines: 50 mg/mL TGFb neutralising antibody (1D11.16.8), or 5 mg/mL IL-10 neutralising antibody (JES3-9D7) (both from Thermo Fisher Scientific). To inhibit PGE_2_ synthesis, 1 mM indomethacin (Sigma Aldrich) was used. A total of 1 mM 1-methyltryptophan (1-MT, Sigma Aldrich, Germany) was used to inhibit kynurenine production. Concentration of blocking agents was chosen on the basis of a previous experiment [28,30]. All blocking agents were added to the ASC cultures. After 48 h, PHA-activated PBMCs and CTV staining were added to the culture. After 5 days of co-culture, PBMCs were harvested for cytometric analyses.

### 2.11. Statistical Analysis

Data were analysed using GraphPad Prism software version 7 (GraphPad Software, Boston, MA, USA). The Shapiro–Wilk test was used to assess data distribution. Mann–Whitney or Kruskal–Wallis tests (for unpaired samples) were used to compare the ASCs obtained from RA and OA patients. For the analysis of the effect of blocking agents on T-cell proliferation, the Friedman test with Dunn’s multiple comparison tests were used. Probability values less than 0.05 were considered significant.

## 3. Results

### 3.1. Phenotype

The cytometric analysis confirmed that 98% of ASCs from RA and OA patients, expressed CD105, CD73 and CD90 molecules. Less than 1% of these cells showed expression of CD45, CD19, CD14 and HLADR proteins (Figure 1). These results are in agreement with the recommendations of the International Society for Cellular Therapy. CD34, whose expression, according to the aforementioned recommendation, should not be present in more than 2% of the cells, was detected in more than 10% of the ASCs analysed in this study. ASCs from RA patients had a significantly higher expression of this protein compared to cells isolated from OA patients. A schematic presentation of the cytometric analysis (gating strategy) is shown in Appendix A.

### 3.2. ASC Differentiation

In the experimental set-up presented here, ASCs cultured in chondrogenic medium increased the expression of mRNAs encoding proteins that are markers of the chondrogenesis: transcription factor *SoX9*, aggrecan (*ACAN*) and collagen 2a (*COL2A1*). At the same time, ASCs secreted extracellular matrix glycosaminoglycans, which were visualised by alcian blue staining of sections obtained from the cell pellets (Figure 2). Expression of chondrogenesis markers at the mRNA level increased to a similar extent in ASCs from RA and OA patients. There were also no differences in glycosaminoglycan deposition.

Culture of ASCs in osteogenic medium resulted in increased transcription of genes that are markers of osteogenesis—the transcription factors Runx2 (*RUNX2*) and osterix (*OSX*), osteopontin (*OPN*), alkaline phosphatase (*ALPL*) and collagen 1 (*COLA1A*). Significantly lower expression of *ALPL*, *OPN* and *RUNX2* was observed in cells isolated from RA patients but Alizarin Red S staining showed no apparent differences between mineralisation in ASCs from RA and OA patients (Figure 3).

The culture of ASCs in adipogenic medium resulted in increased transcription of genes that are markers of adipogenesis (transcription factors *FAB4*, *PPARG* and *CEBPR*) and in the accumulation of fat droplets in the cytoplasm of the cells (Figure 4). No differences were observed between ASCs from RA and OR patients.

### 3.3. Effect of ASCs on Activated T Cell Proliferation

Under αCD3CD28 stimulation, CD4 lymphocytes proliferated intensively. No statistically significant reduction in the percentage of proliferating cells or the proliferation index was observed in the co-cultures of ASCs with activated purified CD4 lymphocytes. (Figure 5).

A different effect was observed when PBMCs were co-cultured with ASCs. PHA-stimulated PBMC T lymphocytes proliferated vigorously. In the co-culture with ASCs, a statistically significant decrease in the percentage of proliferating PBMCs, as well as a decrease in the PI, was observed (Figure 6). Prestimulation of ASCs with TI did not exacerbate the inhibitory effect of ASCs. No significant changes were observed between the effect exerted by cells obtained from RA and OA patients.

### 3.4. Soluble Factors Involved in Antiproliferative Capacities of ASCs

The production of substances potentially related to the inhibition of T lymphocyte proliferation by ASCs was subsequently analysed in the culture supernatants. Concentrations of IL-10, TGFβ, kynurenines and prostaglandin E2 (PGE_2_) were assessed. Significantly increased IL-10 production was observed in the co-culture of ASCs with activated T cells. ASCs cultured separately did not produce IL-10. There was no increase in the production of TGFβ in the co-culture. ASCs produced certain amounts of TGFβ but no increase was shown after TI stimulation. No differences were observed between ASCs from RA and OA patients. After TI stimulation, ASCs produced significant amounts of kynurenine with no differences in the secretion of this substance between cells isolated from RA and OA patients. Higher concentrations of kynurenines were detected in the supernatants from ASCs co-cultured with T cells. Pre-stimulation of ASCs with TI resulted in increased levels of kynurenine production. In all culture options used, there were no differences between cells from OA and RA patients. ASCs spontaneously secreted certain amounts of PGE_2_ but no increase was shown after TI stimulation. In the co-cultures with T cells, a significant increase in secreted PGE_2_ was observed (Figure 7).

No significantly increased IL-10 or TGFβ production was observed in the co-culture of ASCs with activated PBMCs. Increased concentrations of kynurenines were detected in the supernatants from ASCs co-cultured with PBMCs. In the co-cultures with PBMCs, a significant increase in secreted PGE_2_ was observed. Furthermore, a significantly lower production of PGE_2_ was also observed in the co-cultures with RA-ASCs than in co-cultures with OA-ASCs (Figure 8).

### 3.5. Blocking Experiments

To confirm the involvement of soluble factors in inhibiting the proliferation of activated T lymphocytes, blocking experiments were performed. As inhibition of proliferation was only observed in ASC cultures with PBMCs, blocking experiments were performed only for this type of co-culture. PGE_2_ inhibitor, indomethacin and IL-10-neutralising agent restored proliferation of PBMCs co-cultured with OA-ASCs. In co-cultures with OA-ASCs pre-stimulated with TI, inhibition of kynurenines with 1-MT resulted in increased proliferation of PBMCs (Figure 9).

In all co-cultures with RA-ASCs, blocking of PGE_2_ and kynurenines production restored PBMCs’ proliferation. Neutralisation of IL-10 resulted in weak increase of PBMC proliferation, whereas TGFβ blocking had no effect on PBMC proliferation (Figure 10).

## 4. Discussion

In the present study, ASCs derived from the infrapatellar fat pad isolated from the knee joint of RA and OA patients were compared. Aspects considered in this comparison were the phenotype of the cells, their chondrogenic, osteoblastic and adipogenic potential and their antiproliferative properties against T lymphocytes and PBMCs.

Literature data provide extensive information on the regenerative and immunosuppressive properties of mesenchymal stem cells derived from adipose tissue as well as from other tissues [31,32,33,34,35]. Our study is an original and important contribution to the field, as it concerns cells derived from RA and OA patients and thus provides knowledge on how cells function in an inflammatory environment. Moreover, the cells studied were isolated from the infrapatellar fat pad, a fatty body located in close proximity to the synovial membrane. The IFP and the synovium are considered to be in constant contact with each other [36]. Furthermore, the synovium and the IFP are believed to be a single anatomo-functional unit [37]. In the course of the inflammatory process, either in RA or OA, both of these structures are the site of active inflammation.

RA is a systemic autoimmune disease in which many cell types contribute to the development and maintenance of local and generalised high-grade inflammation and autoimmunity. It can be speculated that in the rheumatoid joint, in addition to fibroblastic synoviocytes, autoreactive T and B lymphocytes and other over-activated immune cells, such as mesenchymal stem cells, also have altered functions.

OA, which was treated as a control group in this study, is not an autoimmune disease, although it too is associated with the development of low-grade inflammation. The study shows that MSCs from the infrapatellar fat pad of RA and OA patients are functionally altered, but not significantly different, despite the different pathogenesis of the two diseases. It appears that the micro-inflammatory environment in which these cells are found may be the cause of their functional abnormalities.

Adipose tissue is a readily available and abundant source of mesenchymal stem cells. Compared to the originally characterised mesenchymal stem cells from bone marrow (BM-MSCS), ASCs are characterised by higher expression of stem cell-specific markers and greater resistance to apoptosis [38,39,40]. In addition to their ability to differentiate into various tissues, they are equipped with immunomodulatory properties—they are able to inhibit the activation of T and B lymphocytes and NK cells [41]. The immunomodulatory effect of ASC has also been shown to be stronger than that of BM-MSC [42].

Most commonly, ASCc are isolated from subcutaneous adipose tissue. However, it has been shown that the IFP can be an attractive source of ASCs due to the unique properties of these cells. IFP-ASCs are distinguished by the fact that their degree of proliferation as well as their ability to differentiate are independent of the age of the donor [43]. Furthermore, IFP-ASCs showed increased chondrogenic potential, which is very important in the context treatment of rheumatic diseases [44].

In the first part of this paper, results are presented on the phenotype of ASCs of the infrapatellar fat pad. So far, no specific marker or group of markers for adipose tissue mesenchymal stem cells has been described in addition to the surface proteins CD105, CD90 and CD73, which the International Society for Cell Therapy recommends should be present on the surface of MSCs [45]. Cytometric analysis has shown that ASCs from RA patients, as well as those from OA patients, have a mesenchymal stem cell phenotype. They do not have leukocyte markers (CD14, CD19, CD45) on their surface, but do express CD105, CD90 and CD73. According to literature data [15,16] the analysed cells have a non-immunogenic phenotype, as they have a low expression of HLA-DR molecules belonging to MHC class II. There are also conflicting data that show an increase in HLA-DR expression on OA-ASCs isolated from the infrapatellar fat pad. The authors link this phenotype to a pro-inflammatory joint environment [46]. The CD34 protein, which may be both a marker of haematopoietic cells and vascular endothelial cells, was present in more than 10% of the ASCs analysed in this study. ASCs from RA patients had significantly higher expression of this protein compared to cells isolated from OA patients. This may signal that, in accordance with the observations of other authors [47], we are dealing here with two subpopulations of mesenchymal cells, which differ in terms of CD34 expression, but have similar biological properties. Other authors link unusual expression of CD34 on ASCs with priming by a pathological, proinflammatory environment [46].

It is known that mesenchymal adipose tissue cells have regenerative properties and can differentiate into chondrocytes, osteoblasts and adipocytes in vitro and in vivo. It has been shown that ASCs from the infrapatellar fat pad of RA and OA patients retain the capacity for chondrogenesis and osteogenesis and their differentiation capabilities are comparable [27,48]. Although an increase in the mRNA level of the markers encoding osteogenesis was more marked in cells from OA patients, Alizarin Red S staining showed no apparent differences in mineralisation between cells from RA and OA patients. It therefore appears that also the osteogenic potential of ASCS cells does not differ between the two diseases. Literature data comparing the chondrogenic, adipogenic and osteogenic potential of bone marrow-derived MSSCs from RA and OA patients are similar to those obtained in the present study [49,50]. There is a study indicating impaired chondrogenesis in OA-derived infrapatellar fat pad-derived mesenchymal stem cells [51]. In contrast, other authors have also shown that ASCs isolated from intra-articular adipose tissue of OA patients differentiate into chondrocytes, reduce cartilage degradation and modulate the inflammatory response [52]. These positive results from in vitro experiments with ASCs from patients with OA are already supported by clinical trials [53].

In experiments assessing the proliferation of activated T lymphocytes, no significant inhibition of activated T cell proliferation was observed in T cell–ASCs co-culture. This result is in contrast to previously published observations from studies showing a clear antiproliferative effect of ASCs from adipose tissue of rheumatic patients [28,29,30]. It seems that ASCs from the infrapatellar fat pad are not able to exert a direct inhibitory effect on proliferating T cells. A different effect was observed in PBMC co-cultures with ASCs. In this case, a significant inhibition of T lymphocyte proliferation was observed for both cells obtained from RA and OA patients. ASCs secrete a broad spectrum of cytokines, growth factors and other compounds [54], which have been attributed to immunosuppressive effects against various types of immune cells. Our previous experiments suggest that the antiproliferative effect of ASCs is related to soluble factors and not to the contact between ASC cells and T cells/PBMCs [28,29,30]. Interestingly, an increase in kynurenine and PGE_2_ factors was observed in both types of co-cultures. Further blocking experiments confirmed that inhibition of proliferation is linked to the kynurenine and PGE_2_ pathways. It is likely that in ASC co-culture with PBMCs, additional cells, probably monocytes, contribute to the antiproliferative effect. A similar pathway was shown for Treg generation in ASCs/PBMC co-culture [55].

Observations by other authors also suggest that pro-inflammatory factors present in the joint fluid of RA or OA patients enhance the immunosuppressive properties of MSCs [56]; however, this has not been confirmed in our experiments. The literature emphasises the fact that stimulation with pro-inflammatory cytokines such as IFNγ, TNF, IL-1β or IL-6 either enables or enhances the immunosuppressive function of MSCs from healthy donors [54,57]. It seems that ASCs derived from intra-articular adipose tissue from RA and OA patients, due to their localisation in a persistently inflammatory environment, respond differently to these cytokines than cells from healthy individuals. A published study showing that ASCs isolated from the synovial membrane of RA patients lose their ability to inhibit the proliferation of activated T lymphocytes under the influence of TNF or IL-17A seems to confirm this hypothesis [58]. Impaired inhibition of lymphocyte activation by joint-localised ASCs implies poorer protection against excessive immune responses as well as against potential autoimmune processes.

The idea of using tissue that is waste from a knee replacement surgery as a therapeutic agent is very rewarding. ASCs derived from the infrapatellar fat pad (IFP) have shown promising potential for clinical use in various conditions, including OA and RA, due to their regenerative and immunomodulatory properties. ASCs derived from the IFP can differentiate into chondrocytes or osteoblasts and can promote cartilage or bone regeneration. They also have anti-inflammatory and immunomodulatory properties, which can help reduce joint inflammation and slow down the progression of OA and RA. ASCs derived from the IFP can regulate the immune system by inhibiting the proliferation and activation of immune cells, such as T cells and B cells. This property makes them a potential therapeutic option for RA, which is an autoimmune disease characterised, among others, by pathogenic T cell expansion.

The biggest limitation of our study was that we did not have access to ASCs isolated from healthy IFP. It is not ethically possible to collect such tissue so we could only compare cells taken from OA and RA patients.

## 5. Conclusions

Our experiments showed that ASCs from RA and OA patients have similar differentiation potential and a similar ability to inhibit activated T cells. However, their phenotype and inability to inhibit the proliferation of pure activated T cells appears to be altered by the pro-inflammatory environment from which they originate. Our study revealed that the IFP ASCs of rheumatic patients have shown promising potential for clinical use in various conditions due to their regenerative and immunomodulatory properties. However, more studies are needed to establish their safety and efficacy for these applications.

## Figures and Tables

**Figure 1 pharmaceutics-15-01003-f001:**
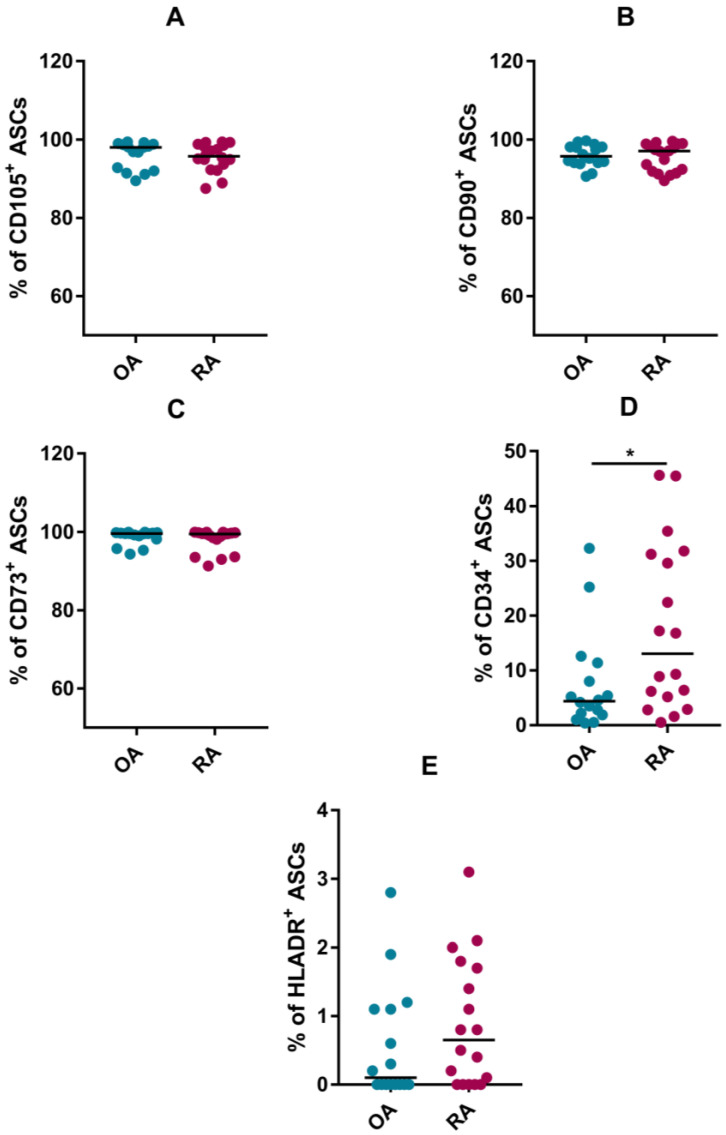
Phenotype of ASCs from RA and OA patients. Plots demonstrate the percentages of RA- and OA-ASCs expressing a given surface protein: CD105 (**A**), CD90 (**B**), CD73 (**C**), CD34 (**D**), HLADR (**E**). Results are shown as single values (dots) and medians (line). A *p* value is expressed as follows: 0.05 > *p* > 0.01 as *.

**Figure 2 pharmaceutics-15-01003-f002:**
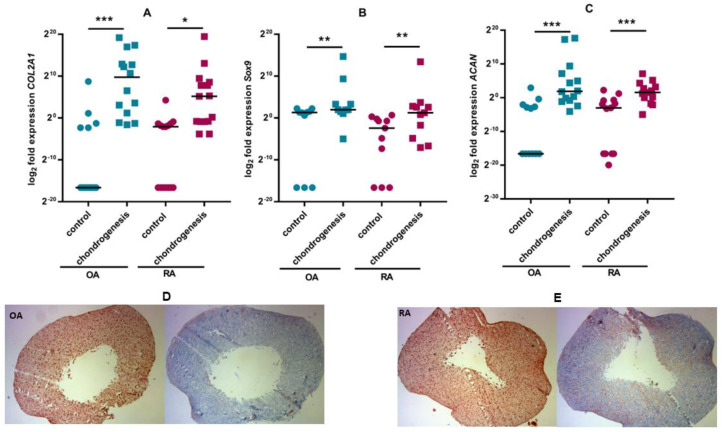
Chondrogenic differentiation of ASCs. The expression of selected chondrogenesis markers at the mRNA level in ASCs cultured in DMEM/F12 medium and in chondrogenic medium was assessed by RT–qPCR (**A**–**C**). Alcian blue staining of pellet scrapings showing deposition of extracellular matrix glycosaminoglycans. The photograph shows the representative staining results of ASCs from OA patients (**D**) and RA patients (**E**). Individual values (circles and squares) and medians a (black line) are shown in the figure. *p* values are expressed as follows: 0.05 > *p* > 0.01 as *; 0.01 > *p* > 0.001 as **; *p* < 0.001 as ***.

**Figure 3 pharmaceutics-15-01003-f003:**
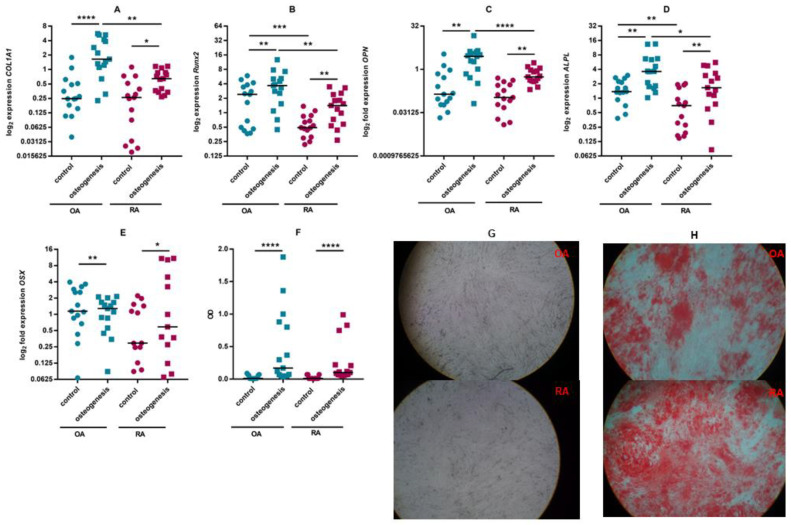
Differentiation of ASCs towards osteoblasts. The expression of selected osteogenesis markers at the mRNA level in cells cultured in DMEM/F12 medium and in osteogenic medium was assessed by RT-qPCR (**A**–**E**). Staining of cells with Alizarin Red S highlighting calcium deposition. The red dye was extracted and its concentration assessed spectrophotometrically (**F**). The photographs show representative cultures of unstained ASCs (**G**) and ASCs stained with Aizarin Red S (**H**) after osteogenic differentiation. Individual values (circles and squares) and medians (black line) are shown in the figure. *p* values are expressed as follows: 0.05 > *p* > 0.01 as *; 0.01 > *p* > 0.001 as **; *p* < 0.001 as ***; *p* < 0.0001 as ****.

**Figure 4 pharmaceutics-15-01003-f004:**
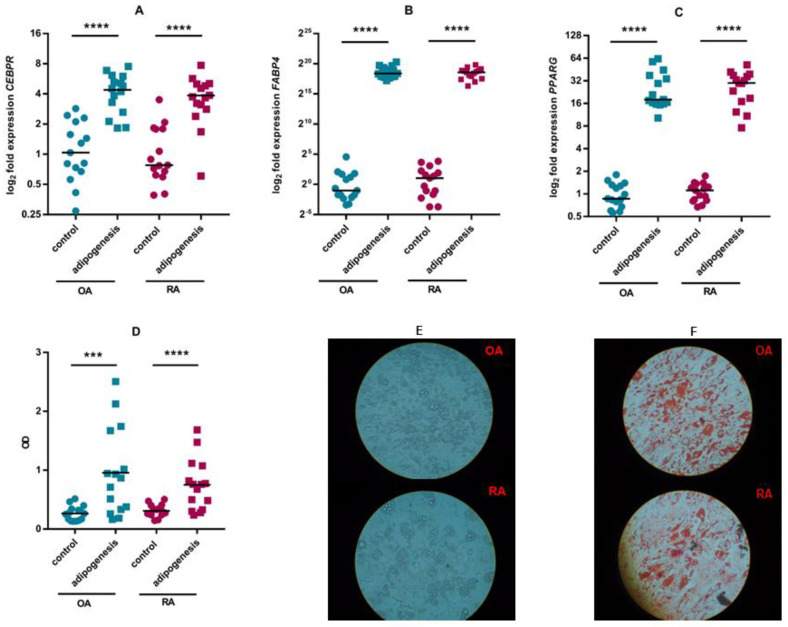
Differentiation of ASCs towards adipocytes. The expression of selected adipogenesis markers at the mRNA level in cells cultured in DMEM/F12 medium and in adipogenic medium was assessed by RT–qPCR (**A**–**C**). Fat droplets in the cytoplasm were stained using Oil Red O. The red dye was extracted and its concentration was assessed spectrophotometrically (**D**). The photographs show representative ASC cultures before (**E**) and after staining with Oil Red O (**F**). Individual values (circles and squares) and medians (black line) are shown in the figure. *p* values are expressed as follows: *p* < 0.001 as ***; *p* < 0.0001 as ****.

**Figure 5 pharmaceutics-15-01003-f005:**
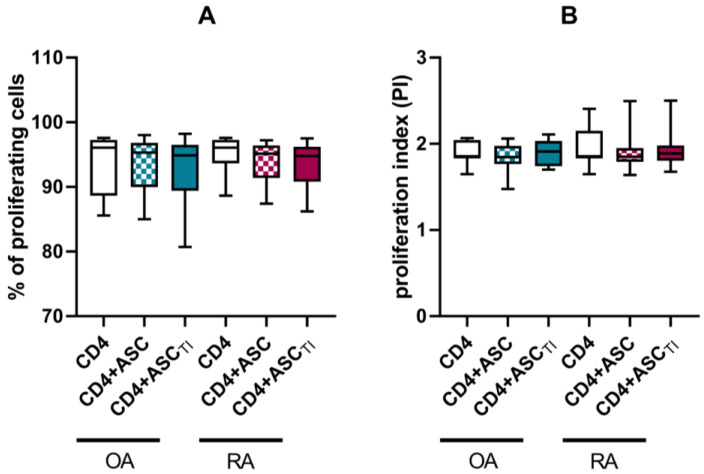
Proliferation of T cells co-cultured with ASCs of OA patients or RA patients. CD3+CD4+ cells were isolated from PBMCs obtained from healthy donors and stimulated with αCD3CD28. CD3+CD4+ were cultured alone (control) or co-cultured for 5 days with either untreated or TI-stimulated ASCs. The proliferation was assessed as percentage of proliferating cells (**A**) or proliferation index (**B**).

**Figure 6 pharmaceutics-15-01003-f006:**
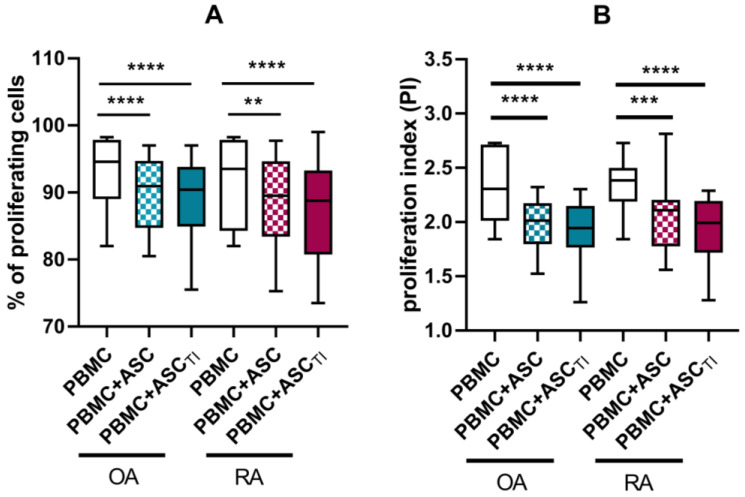
Proliferation of PMBCs co-cultured with ASCs of OA patients or RA patients. PBMCs obtained from healthy donors were stimulated with PHA. PBMCs were cultured alone (control) or co-cultured for 5 days with either untreated or TI-stimulated ASCs. The proliferation was assessed as percentage of proliferating cells (**A**) or proliferation index (**B**). *p* values are expressed as follows: 0.01 > *p* > 0.001 as **; *p* < 0.001 as ***; *p* < 0.0001 as ****.

**Figure 7 pharmaceutics-15-01003-f007:**
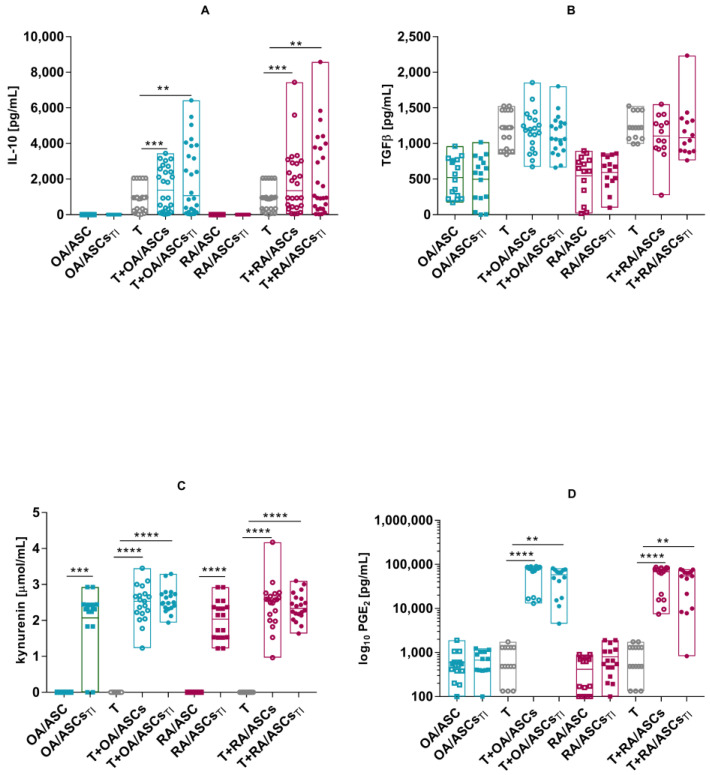
Concentrations of IL-10 (**A**), TGFβ (**B**), kynurenine (**C**) and PGE_2_ (**D**) in the co-cultures of OA- (teal signs) or RA-ASCs (red signs) with αCD3CD28-activated T cells. Data are shown as single values with box representing 25th and 75th percentiles and medians as lines. *p* values are expressed as follows: 0.01 > *p* > 0.001 as **; *p* < 0.001 as ***; *p* < 0.0001 as ****.

**Figure 8 pharmaceutics-15-01003-f008:**
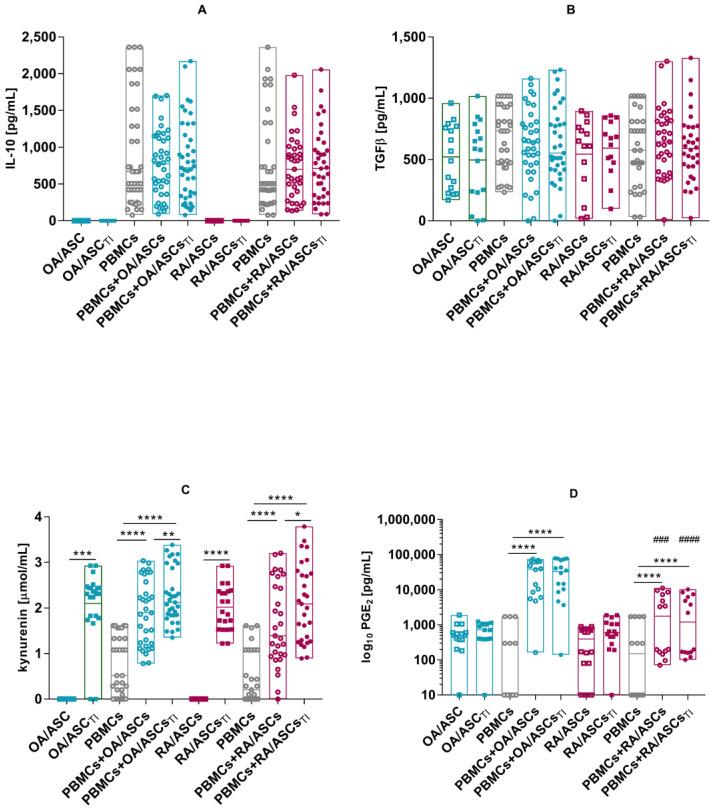
Concentrations of IL-10 (**A**), TGFβ (**B**), kynurenine (**C**) and PGE_2_ (**D**) in the co-cultures of OA- (teal signs) or RA-ASCs (red signs) with PHA-activated PBMCs. Data are shown as single values with box representing 25th and 75th percentiles and, medians as lines. *p* values are expressed as follows: 0.05 > *p* > 0.01 as *; 0.01 > *p* > 0.001 as **; *p* < 0.001 as ***; *p* < 0.0001 as **** for comparisons of cell co-cultures versus separate control cultures. *p* < 0.001 as ###; *p* < 0.0001 as #### for the groups of OA patients versus RA patients comparison.

**Figure 9 pharmaceutics-15-01003-f009:**
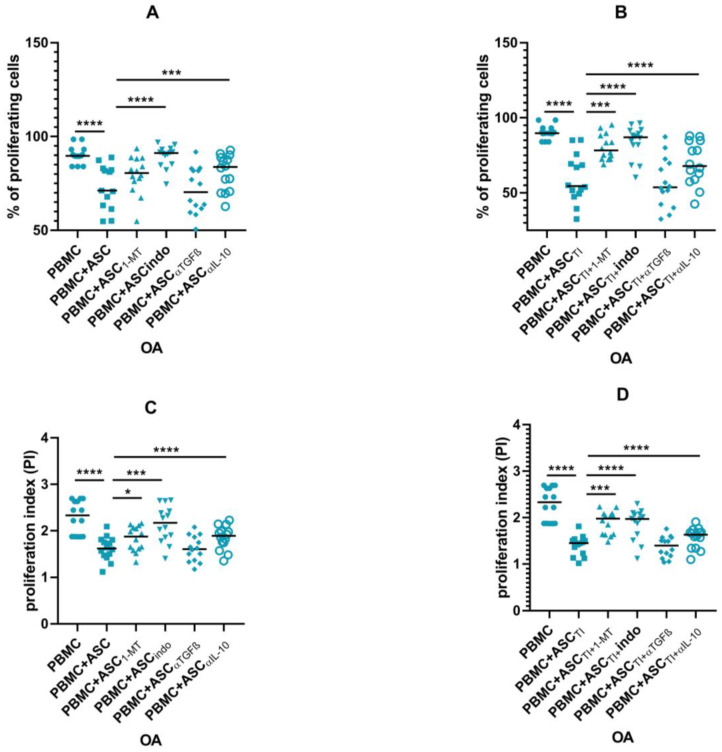
Effects of selective inhibition of kynurenines (with 1-MT), PGE_2_ (with indomethacin), IL-10 neutralisation or TGFβ neutralisation on proliferation of PBMCs co-cultured with OA-ASCs. The proliferation is presented as percentage of proliferating cells (**A**,**B**) or proliferation index (**C**,**D**). *p* values are expressed as follows: 0.05 > *p* > 0.01 as *; *p* < 0.001 as ***; *p* < 0.0001 as ****.

**Figure 10 pharmaceutics-15-01003-f010:**
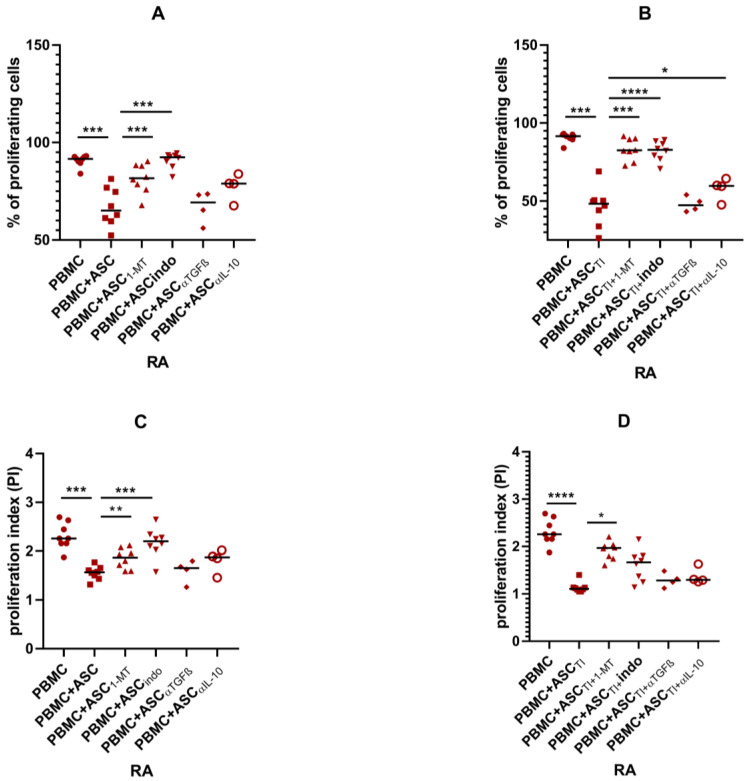
Effects of selective inhibition of kynurenines (with 1-MT), PGE_2_ (with indomethacin), IL-10 neutralisation or TGFβ neutralisation on proliferation of PBMCs co-cultured with RA-ASCs. The proliferation is presented as percentage of proliferating cells (**A**,**B**) or proliferation index (**C**,**D**). *p* values are expressed as follows: 0.05 > *p* > 0.01 as *; 0.01 > *p* > 0.001 as **; *p* < 0.001 as ***; *p* < 0.0001 as ****.

**Table 1 pharmaceutics-15-01003-t001:** Demographic and clinical characteristics of the patients.

	Rheumatoid Arthritis (RA) (*n* = 18)	Osteoarthritis (OA) (*n* = 16)
Demographics		
Age, years	61 (44–69)	64 (39–69)
Sex, female (F)/male (M), *n*	12 F/6 M	10 F/6 M
BMI, kg/m^2^	28.4 (23.3–37.5)	34.2 (27.7–45.5) ^#^
Laboratory values		
CRP, mg/L	13.2 (2–36)	4 (1–14) ^###^
ESR, mm/h	19 (5–77)	14 (2–32) ^#^
Treatment, %		
Methotrexate	55%	0%
Glucocorticoids	66%	0%
Sulfasalazine	33%	0%
NSAIDs	66%	81%

Except where indicated otherwise, values are the median (min–max). BMI, body mass index; CRP, C-reactive protein; ESR, erythrocyte sedimentation rate; NSAIDs, Non-steroidal anti-inflammatory drugs. ^#^
*p* = 0.05–0.01, and ^###^
*p* = 0.01–0.001 for RA vs. OAc comparisons.

**Table 2 pharmaceutics-15-01003-t002:** List of analysed genes.

Gene Symbol	Assay ID	Gene Name
Markers of adipogenesis		
*CEBPB*	Hs00942496_s1	CCAAAT/enhancer-binding protein beta
*FABP4*	Hs01086177_m1	Fatty acid binding protein 4
*PPARG*	Hs01115513_m1	Peroxisome proliferator-activated receptor gamma
Markers of osteogenesis		
*RUNX2*	Hs01047973_m1	Runt-related transcription factor 2
*OPN*	Hs00959010_m1	Osteopontin
*COL1A1*	Hs00164004_m1	Collagen type I α 1
*OSX*	Hs01866874_s1	osterix
*ALPL*	Hs01029144_m1	Alkaline phosphatase
Markers of chondrogenesis		
*SOX9*	Hs00165814_m1	SRY (sex determining region Y)-box 9
*ACAN*	Hs00153936_m1	Aggrecan
*COL2A1*	Hs00264051_m1	Collagen type II α 1
Reference gene		
*GUSB*	Hs00939627_m1	Glucuronidase β
*RPL13A*	Hs04194366_g1	Ribosomal protein L13a
*TBP*	Hs00427620_m1	TATA-binding protein
*eEF-1*	Hs02339452_g1	eukaryotic elongation factor 1 gamma-like protein

## Data Availability

The data presented in this study are available on request from the corresponding author.

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
