# Peer review of "Basic Properties of Adipose-Derived Mesenchymal Stem Cells of Rheumatoid Arthritis and Osteoarthritis Patients"

_pharmaceutics, 2023, doi:10.3390/pharmaceutics15031003_

Round 1
Reviewer 1 Report
Thank you for giving me this opportunity to reivew the research article entitled, "Basic properties of Adipose-Derived Mesenchymal Stem Cells of Rheumatoid Arthritis and Osteoarthritis patients".
I here carefully reviewed the submitted set of the manucript and found it possibly merits for publication after major revisions as mentioned below.
1. In the abstract, the summary seems very rough and unclear for the dreaders, should be amended thoroughly with more concrete and clear descriptions.
2. Table 1 should be revised. Some words and sentences are displaced, and some technical words should be spelled out.
3. In the M&M section, specimens of infrapatellar fat pad should be further well explained and described, whether these are originated from anatomicall localized in the site of ongoing inflammatory process or not. Further, this important issue should be well discussed in the Discussion section as well.
4. All through the article, the cell number as "5 x 104 cells" must be corrected.
5. All the figures are very small and unclear, hard to follow and evaluate, must be revised.
6. In the Discussion section, ASCs derived from infrapatellar fat pad isolated from the knee joint of RA and OA, should be further discussed with approproate referrences. The normal basic properties of Adipose-Derived Mesenchymal Stem Cells of healthy subjects need to be compared and discussed in terms of potentials of the same regenerative and immunologcial biological behaviors.
7. The results obtained in the present study can be how and with what way interpreted and applied into the clinical issues and the treatment. These points should be disccused.
Reviewer 2 Report
The study entitled “Basic properties of Adipose-Derived Mesenchymal Stem Cells of Rheumatoid Arthritis and Osteoarthritis patients” aims at investigating the phenotype, regenerative potential, and effect of adipose derived mesenchymal stem cells, obtained from rheumatoid arthritis and osteoarthritis patients, on T cell proliferation.
Although this is an area of considerable interest, there are several points that need improvement to make this paper suitable for the readership of Pharmaceutics.
Abstract
Line 17: Please add “stem” after “mesenchymal.”
Introduction
Because of the absence of the vascularization in cartilage, chondrocytes are continuously exposed to a hypoxic environment. Therefore, since oxygen tension has been suggested as an important regulatory factor for chondrogenic differentiation of stem cells, the authors should discuss this topic in the Introduction section, especially focusing on the fact that proliferation and differentiation of ASCs depend on their niche (microenvironment) and substrate. The following paper may help the authors to better support their sentences:
· Oztürk E, Hobiger S, Despot-Slade E, Pichler M, Zenobi-Wong M. Hypoxia regulates RhoA and Wnt/beta-catenin signaling in a context-dependent way to control redifferentiation of chondrocytes. Sci Rep 2017;7(1):9032. https://doi.org/10.1038/s41598-017-09505-6.
· Antebi, B., Rodriguez, L.A., Walker, K.P. et al. Short-term physiological hypoxia potentiates the therapeutic function of mesenchymal stem cells. Stem Cell Res Ther 9, 265 (2018). https://doi.org/10.1186/s13287-018-1007-x
· Govoni M., Muscari C., Bonafè F., Morselli P.G., Cortesi M., Dallari D., Giordano E. A brief very-low oxygen tension regimen is sufficient for the early chondrogenic commitment of human adipose-derived mesenchymal stem cells. Adv Med Sci. 2021 Mar;66(1):98-104. doi: https://doi.org/10.1016/j.advms.2020.12.005
Lines 63-67: When the authors assert that “encouraging results of autologous or allogeneic MSCs transplants have been obtained in Crohn’s disease, cirrhosis, heart infarction, graft versus host disease (GvHD), osteogenesis imperfecta, systemic lupus erythematosus (SLE), critical limb ischemia, osteoarthritis and in the reconstruction of bone, cartilage or soft tissues”, an appropriate reference should be added for each disease listed in the sentence.
Lines 69-71: Although the authors give a hypothesis on the regenerative and immunomodulatory properties of ASCs localized in the rheumatoid joint, the aim of this study should be better focused and described.
Materials and Methods
2.2 ASC Isolation and Culture
Authors report that ASC isolation and culture were previously described in the Ref. #21 (Skalska, U.; Kuca-Warnawin, E.; Kornatka, A.; Janicka, I.; Musiałowicz, U.; Burakowski, T.; Kontny, E. Articular and 545 subcutaneous adipose tissues of rheumatoid arthritis patients represent equal sources of immunoregulatory mesenchymal stem 546 cells. Autoimmunity 2017, 50, 441-450).
However, please note that this sentence is not correct.
ASC isolation and culture were described in the following reference of the same research group: Skalska U, Kontny E, Prochorec-Sobieszek M, Maśliński W. Intra-articular adipose-derived mesenchymal stem cells from rheumatoid arthritis patients maintain the function of chondrogenic differentiation. Rheumatology (Oxford). 2012 Oct;51(10):1757-64. doi: 10.1093/rheumatology/kes129.
Therefore, please add in the main text the correct reference.
Line 98: Replace 5 x 104 cells with 5 x 104 cells.
Line 108: Replace 35 x 105 with 35 x 105.
Line 130: Replace 11 x 105 with 11 x 105.
Line 152: Replace 0.5-1 x 106 with 0.5-1 x 106.
2.7 Gene expression analysis
This paragraph should be slightly improved:
- Please, add the amount of total RNA used to perform the reverse-transcription.
- Please, give information about the software used to perform data analysis.
- Please, cDNA instead RT product.
This reviewer strongly recommends following the MIQE guidelines (The MIQE guidelines: minimum information for publication of quantitative real-time PCR experiments. Clin Chem. 2009 Apr;55(4):611-22. doi: 10.1373/clinchem.2008.112797) to use the correct nomenclature in the text. In this regard, please use the term reference gene instead of house-keeping gene, and use the acronym RT-qPCR instead of qRT-PCR in both the main text and figure legends.
Has the M-value (the measure of expression stability) of your reference genes been calculated?
Throughout the manuscript, please refer to the following reference for the correct nomenclature to use for human genes, especially for the use of italics:
Bruford, E. A., Braschi, B., Denny, P., Jones, T., Seal, R. L., & Tweedie, S. (2020). Guidelines for human gene nomenclature. Nature genetics, 52(8), 754–758. https://doi.org/10.1038/s41588-020-0669-3
Line 195: Please replace 6x104 with 6x104.
Line 206: Please replace 1.2 × 106 with 1.2 x 106.
Results
3.2 ASC differentiation
Why have authors decided to use Log10 instead of more common Log2 for the normalization of gene expression?
Figures 2A-E, 3A-H, 4A-F are too small. Is it not possible enlarge them without loosening resolution?
3.3 Effect of ASCs on activated T cells proliferation
Figure 5 and 6.
With the type of graph chosen by authors, it is very difficult to appreciate the effect of ASCs on activated T cells proliferation both in terms of percentage of proliferating cells and proliferation index.
Therefore, this reviewer suggests showing data in a different way.
Discussion
Lines 393-395: Since authors assert that “Literature data provide extensive information on the regenerative and immunosuppressive properties of mesenchymal stem cells derived from adipose tissue as well as from other tissues” more appropriate references should be cited to support this sentence. In this respect, please note that the only reference insert in the present manuscript (Ref#25: Tuan et al.) is older than 20 years.
Lines 405-406: The rationale behind the choice to use OA patients as the control group should be better discussed by the authors providing also appropriate reference to support it.
Moreover, limitations of this study should be better listed and commented.
Minor points
Although the authors have referred in several parts of the manuscript (lines 103-104, 212-213, 258, 482-484) to Figures S1 and S2, for this reviewer was not possible downloading Supplementary Materials.
Round 2
Reviewer 1 Report
I carefully re-reviewed the submitted set of the manuscript and found it merits for publication in the present revised form.
Reviewer 2 Report
Although M-value was not calculated, authors have ameliorated the quality of the manuscript.
Best regards.